# Modelling alcohol consumption patterns to enable policy impact assessment

Jasper ten Dam[1,2]*, A. Jeroen Rodenburg[1], Hendrik Koffijberg[2], Talitha L. Feenstra[3,4], Anoukh van Giessen[2]

1 Department of Statistics Data Science and Modelling, National Institute for Public Health and the Environment (RIVM), Bilthoven, The Netherlands, 2 Health Technology and Services Research Department, Technical Medical Centre, University of Twente, Enschede, The Netherlands, 3 Department of Epidemiology, University Medical Centre Groningen, Groningen, The Netherlands, 4 Centre for Public Health, Healthcare and Society, National Institute for Public Health and the Environment (RIVM), Bilthoven, The Netherlands

* jasper.ten.dam@rivm.nl

## Abstract

### Objective

To prevent harmful effects of alcohol use, various countries implement policies preventing excessive and heavy episodic drinking. To enable the evaluation of the impact of such policies on (future) drinking behaviour, we aimed to develop a model that predicts alcohol consumption patterns.

### Methods

The model predicts alcohol use in three stages. First, a logistic submodel predicts probabilities of drinking any alcohol. Second, for drinkers, a submodel predicts the weekly consumption through a negative binomial distribution for the number of beverages. Finally, based on the predicted weekly consumption, a logistic submodel predicts probabilities of heavy episodic drinking. The distribution for the weekly consumption was calibrated, targeted to predict the prevalence of excessive and heavy episodic drinking accurately. Model parameters were estimated using Dutch individual-level cross-sectional survey data covering the years 2008–2022. The characteristics age, sex, education, calendar time and their interactions were used as predictors and the model accounts for trend breaks in the data. Model performance was assessed by comparing population-level predictions with observed data on which the model was calibrated (2014–2022).

### Results

A comparison between predictions of the calibrated model and observed data shows that the prevalences of excessive (error <0.2 percent point (pp)) and heavy episodic drinking (error <0.1 pp) align, averaged over the years 2014–2022. Visual inspection

**Data availability statement:** The National Health Survey data that support the findings of this study cannot be shared publicly because it is third party data. Access can be requested through Data Archiving and Network Services

(DANS) Data Station Life Sciences (contact via info@dans.knaw.nl) for the following year-specific DOIs: 2008-2010: https://doi.org/10.17026/dans-zrm-7r4z; 2010, 2011: https://doi.org/10.17026/dans-z93-mj8s; 2012: https://doi.org/10.17026/dans-zcc-5stc; 2013: https://doi.org/10.17026/dans-zdk-dwmn; 2014: https://doi.org/10.17026/dans-xcm-u69z; 2015: https://doi.org/10.17026/dans-xwr-m26w; 2016: https://doi.org/10.17026/dans-xxa-e3m7; 2017: https://doi.org/10.17026/dans-xxd-j335; 2018: https://doi.org/10.17026/dans-z5s-b7ve; 2019: https://doi.org/10.17026/dans-xjc-xnf9; 2020: https://doi.org/10.17026/dans-x58-3ayy; 2021: https://doi.org/10.17026/dans-x2a-yw93; 2022: https://doi.org/10.17026/LS/0OKUIT.

**Funding:** This study was funded by the Dutch Ministry of Health, Welfare and Sport and the National Institute for Public Health and Environment (RIVM).

**Competing interests:** The authors have declared that no competing interests exist.

using qq-plots and within-sample validation over time further indicates that the model fits well for predicting excessive and heavy episodic drinking, based on the predicted distribution for the weekly consumption.

## Conclusions

We developed a model for alcohol consumption patterns based on Dutch data. This model enables evaluation of the impact of interventions on the (future) prevalence of excessive and heavy episodic drinking.

## Introduction

There are growing health concerns surrounding excessive alcohol use, as it is one of the leading causes of preventable disease burden and mortality [1]. For example, in the Netherlands, it is estimated that 3% of all disability adjusted life years (DALYs) are attributed alcohol use [2]. Long-term health risks of alcohol use include the development of chronic diseases and other serious health risks, for example, high blood pressure, weakening of the immune system, alcohol use disorder and mental health problems [3]. The WHO encourages policy-makers to implement strategies that have shown to be effective and cost-effective to reduce harmful alcohol use [1,4,5]. Public health policies could potentially influence quality of life, disease burden and mortality and are showing promising evidence of cost-effectiveness [6,7].

In the Netherlands, for example, public health policies aiming to reduce excessive and heavy episodic drinking are implemented through a collaborative agreement involving the government and numerous parties, such as healthcare and civil society organizations [8]. This agreement contains measures, such as restriction of discounts on alcoholic beverages and a screening program to enable early detection of alcohol-related issues [8]. Additionally it contains goals to decrease the prevalence of excessive drinking among adults from 8.8% in 2018 to 5% in 2040 and of heavy episodic drinking from 8.5% in 2018 to 5% in 2040 [8]. Excessive drinking is defined in the Netherlands as drinking more than 21 (men) or 14 (women) alcoholic beverages per week. Heavy episodic drinking, or 'heavy drinking', is defined as drinking 6 alcoholic beverages (men) or 4 alcoholic beverages (women) on a day, at least once a week [8,9].

The impact of such policy measures should be evaluated, for instance to assess whether they are sufficient to achieve the goals set for 2040. This requires models to predict future alcohol consumption and alcohol consumption patterns. In public health, predictions are usually based on widely available characteristics, such as age, sex and, possibly, level of education. Additionally, time trends reflect the impact of previous alcohol-related policies and developments (e.g., [10–12]). Hence, a model for alcohol use requires the distribution of use by demography (age, sex, education), as well as time trends of use. Alcohol prevention policies aim to reduce alcohol consumption of individuals, and are commonly evaluated based on population-level prevalences of, for instance, excessive and heavy drinking. Hence, an alcohol model for the evaluation of prevention policies should predict the distribution of alcohol

consumption, as well as (population-level) excessive and heavy drinking prevalences, and include time trends. This model can then be included in health impact analysis models to assess the impact of policy interventions by predicting future alcohol consumption, as well as alcohol related diseases, mortality, costs and quality of life.

Several existing models predict alcohol use for the general population and are applied in policy evaluation [13–18]. Some of these studies modelled alcohol consumption as a categorical variable (e.g., abstention, moderate consumption, heavy consumption) [14,18]. However, the effects of interventions on alcohol use are most often measured through a change in the number of beverages consumed [19–22]. Hence, in a model where alcohol consumption is treated as a categorical variable, any change in the number of beverages should be reflected in the transition probability to another category. Therefore, these models are limited in incorporating interventions. To align an alcohol model with the effect size of interventions, alcohol consumption ideally should be modelled using a continuous or discrete distribution representing the amount of alcohol consumed in more detail. The only identified model that predicts alcohol consumption for a general population as a non-categorical variable is the public health simulation model from the Organization of Economic Collaboration and Development (OECD) [17]. However, this model does not include a time trend and was based on aggregated data (i.e., not on individual-level data).

Our objective was to develop a model for alcohol use, including the distribution of alcohol consumption as well as variables representing excessive and heavy drinking, using individual-level data on alcohol use over time in the Netherlands. When implemented in a simulation model for public health, this model enables the evaluation of future policy measures.

## Materials and methods

### Model structure

The alcohol model predicts different outcomes related to alcohol use in three stages. First, a submodel predicts drinking and not drinking. Second, for drinkers only, a submodel predicts alcohol consumption through the number of alcoholic beverages per week (NABW). From the NABW, excessive drinkers are derived. Third, a submodel predicts heavy drinking. Fig 1 shows a conceptual overview of the alcohol model. The model is designed such that the effects of policy interventions can be specified through adjustment of the NABW.

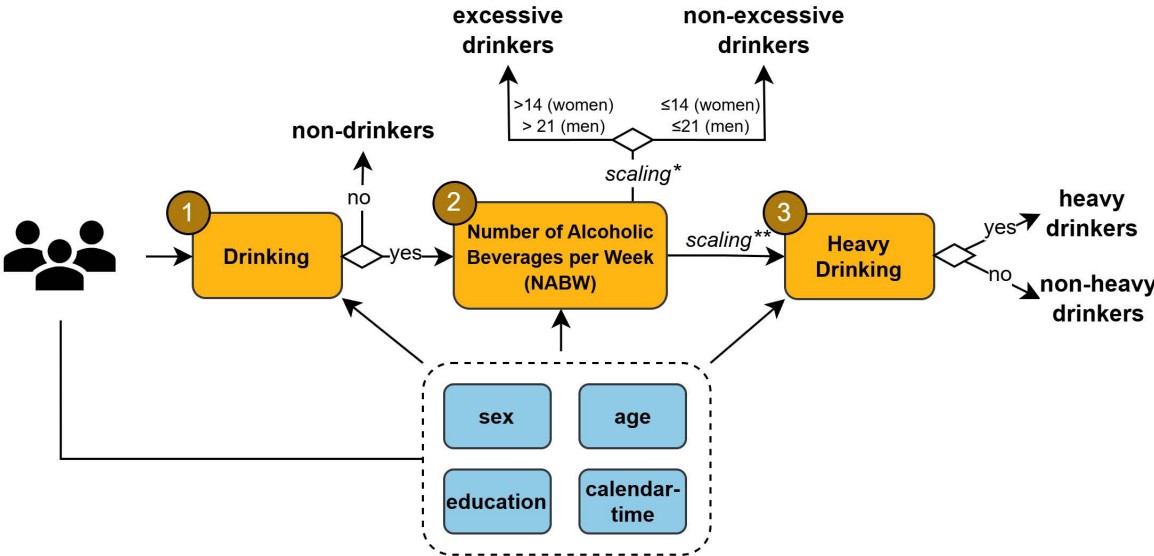

**Fig 1. Conceptual model overview.** Submodels predict three stages of the total alcohol model (1) drinking; and for drinkers: (2) the NABW and excessive drinking, and (3) heavy drinking. *Scaling of the NABW, based on a calibration procedure for excessive drinking. ** Scaling of the NABW, based on a calibration procedure for heavy drinking.

## Data and pre-processing

The total alcohol model was fitted on an annual cross-sectional survey on health and lifestyle: the National Health Survey [23]. The population-level outcomes related to alcohol use from the survey are representative of the Dutch population via the use of supplied weighting variables. Detailed methodological description on the weighing variables is provided in elsewhere by Statistics Netherlands [24–29]. Next to variables on (patterns of) alcohol use, age (by 1-year), sex (male, female), year (expressed as the decimal year in which a respondent completed the survey, further denoted as calendar time), and education (low, middle, high) were extracted from the dataset. For education, the lowest level aggregates people with intermediate secondary education or less and basic vocational education, the medium level aggregates higher secondary and intermediate vocational education, and the highest level applies to persons with higher vocational education or university. Although the model predicts alcohol consumption for adults (18+), data from youth aged 12–18 years were also included in model parameter estimates to improve the model fit for young adults. Education was defined as the (self-reported) highest completed level of education. The dataset encompasses 107,535 individuals over 15 years, ranging between 5,500 and 8,500 per year.

Alcohol use was assessed based on several questions (see Table A in S1 File). These questions regard whether the person drank alcohol in the last year, the frequency of alcohol use and the amount of alcohol that has been consumed on a typical day. Within a typical week, latter two items were determined separately for weekdays and weekend days. Hence, the number of beverages on weekdays and weekend days were calculated by multiplying the number of days on which a person drinks alcohol with the average amount of alcohol that is typically consumed on such a day. The amount of alcohol was assessed in the survey by asking the number of alcoholic beverages on a typical day of consuming alcohol. An alcoholic beverage is assumed to be equivalent to a standard drink, which is a way to ensure comparability across different beverages. The definition of a standard drink varies between countries [30,31]. In the Netherlands, it is a drink that contains 10 grams of pure alcohol [31]. Based on the variables in the data, individual alcohol consumption was calculated and defined through the number of alcoholic beverages per (typical) week (NABW).

A person was regarded a drinker when the NABW was larger than 0 (i.e., when consuming at least one alcoholic beverage in a typical week in the last year), and as a non-drinker otherwise. A person was considered an excessive drinker when the NABW is larger than 14 (women) or 21 (men) [9]. Whether a person is a heavy drinker is determined by questions assessing whether a person drinks at least 6 (men or women before 2012) or 4 (women only from 2012 due to definition change) alcoholic beverages on one occasion at least once a week [9].

The years 2008 until 2022 were chosen to fit the model. This time period was assumed to give the best representation of the time trend in drinking behaviour taking into account the history of alcohol-related policies and external impacts and developments, such as the number of alcohol outlets or alcohol promotion on social media. Table A in S2 File provides an overview of the baseline statistics on demographics and alcohol use of the survey population in the years 2008–2022.

The data shows trend breaks due to redesigns of the survey over time. These redesigns placed alcohol-related questions in a different position in the survey, changed questions or changed a definition in the case of heavy drinking for women (see above). The trend breaks occurred between the years 2009 and 2010 (changed questions and repositioning of questions), 2011 and 2012 (extra question for definition change for heavy drinking for women) and 2013 and 2014 (changed questions and repositioning of questions) [32–34]. The histograms in Fig 2 show the frequency distribution of the NABW for the most recent period without trend breaks (2014 until 2022). All (statistical) analyses were performed in R version 4.4.2, using RStudio [35].

## Submodel 1: drinking or not drinking

The data for the NABW contained many zeros (24.8% of individuals were non-drinkers, see Fig 2). To account for this zero-inflation, the first submodel of the alcohol model reflects the probability of drinking (NABW ≥ 1) or not drinking (NABW = 0). We defined the probability for an individual to drink as $p^d$. Next, we regressed the log odds of this probability

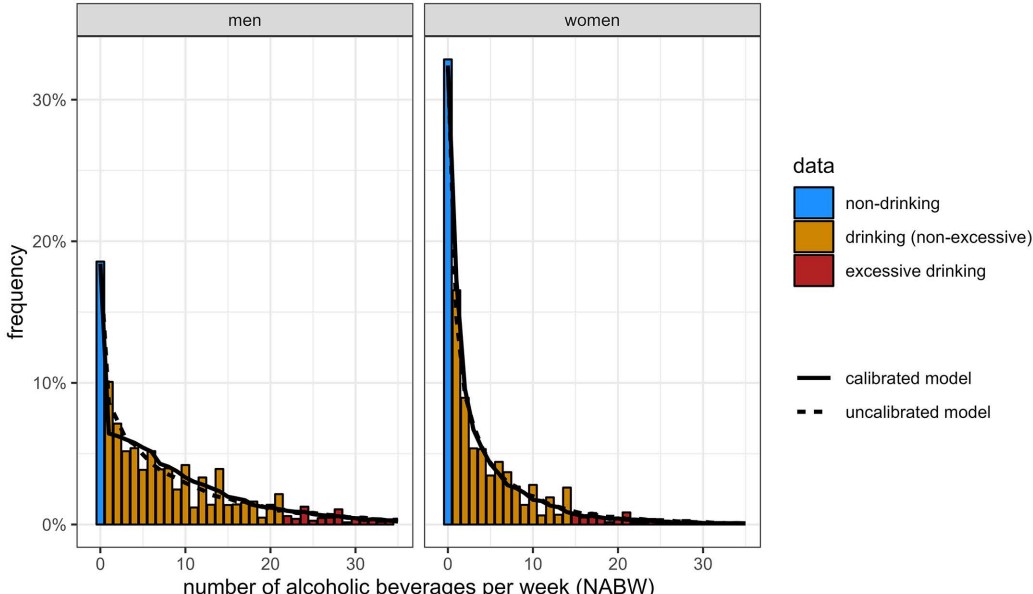

**Fig 2. NABW frequency distributions.** Frequency distributions of the number of beverages per week (NABW) for adult men and women, as predicted by the uncalibrated model, the model calibrated for excessive drinking, and from data, in the calibration period (2014-2022). Data in plot truncated for NABW > 35 (showing >97.5% of the data).

through a logistic model. We parameterised the log odds, for an individual with sex $s$, level of education $e$ and age $a$ at calendar time $t$, as

$$\log\left(\frac{p^d}{1-p^d}\right) = B^{(1)}_{s,e}(a) + B^{(2)}_{s,e}(a) \cdot t,$$

(1)

where $B_{s,e}(a)$ are B-spline functions or ordinary polynomial functions of the individual's age $a$, which were determined separately by sex $s$ and level of education $e$. The term $B^{(2)}$ interacts with calendar time $t$, such that this term describes a time trend in the probability of being a drinker that depends on age, sex and education.

Model selection was based on Akaike information criteria (AIC) for the first term, $B^{(1)}_{s,e}$. All pairwise interactions between sex, age and education were found to result in an improved AIC, compared to not including interaction terms. We set the knots of this B-spline at equal distance and chose the spline order and the number of knots based on AIC, resulting in a piecewise third order polynomial with 5 internal knots. It is important that the term interacting with time, $B^{(2)}_{s,e}$, has an age dependence that is relatively smooth, as the model will be extrapolated to the future. If $B^{(2)}_{s,e}$ fluctuates too much with age, extrapolating this term will cause large fluctuations with age that are probably unrealistic. Therefore, we described $B^{(2)}_{s,e}$ by a (non-piecewise) second-order polynomial. The parameters defining the splines $B^{(1)}_{s,e}$ and $B^{(2)}_{s,e}$ were estimated through logistic regression. We did not adjust this submodel for trend breaks in the data, as the drinking prevalence did not seem to be (largely) affected by them (see Fig A in S3 File).

### Submodel 2: number of weekly beverages and excessive drinking

The second submodel describes the distribution of the NABW for persons drinking at least one alcoholic beverage per week ($NABW \geq 1$). Fig 2 shows that the empirical distribution of the NABW is irregular. This may be a result from the preprocessing step, where the NABW is obtained by adding up integer values for the typical number of beverages on an

answered number of weekdays and weekend days. The outcome of this calculation may be sensitive to preferable answer options, which occurred often in the data (e.g., drinking each week day and/or weekend day). We treated the NABW variable, which consists of integer values, as a count variable and assumed that, for drinking individuals ($NABW \geq 1$), the NABW subtracted by 1 follows a negative binomial (NB) distribution:

$$(NABW - 1) \sim NB(\mu, \theta), \tag{2}$$

where $\mu$ denotes the mean NABW and $\theta$ denotes the dispersion parameter. We fitted the distribution as a generalized linear model [36], separately for men and women. This allowed to estimate the dispersion by sex (i.e., $\theta = \theta_s$). The log of the mean NABW for an individual, $\mu$, was parameterized as a function of the individual's sex $s$, education $e$, age $a$ and calendar time $t$ as follows:

$$\log(\mu) = B_{s,e}^{(3)}(a) + B_{s,e}^{(4)}(a) \cdot t$$

$$+ \gamma_s^1 \cdot H(t - 2010) + \gamma_s^2 \cdot H(t - 2012) + \gamma_s^3 \cdot H(t - 2014), \tag{3}$$

where $B_{s,e}(a)$ are, similar to the first submodel, B-splines or polynomial functions of the individual's age $a$. The second line of the formula defines for men and women three parameters $\gamma_s^{1,2,3}$, which adjust for the trend breaks in the years 2010, 2012 and 2014, respectively. Here, $H$ denotes the Heaviside step function, defined as $H(\tau) = 1$ for $\tau \geq 0$ and $H(\tau) = 0$ for $\tau < 0$. Similar to the first submodel, for the first term, $B_{s,e}^{(3)}$, model selection was based on AIC. A piecewise third order polynomial with four internal knots was selected, as it yielded the lowest the AIC. In line with the first submodel, we limited $B_{s,e}^{(4)}$ to a second order polynomial.

**Calibration.** Excessive drinking is based on a threshold on the NABW distribution. As the ultimate aim of the model is to predict the prevalence of excessive and heavy drinking, it is more important to predict the prevalence of excessive drinking accurately, than the mean NABW. We used qq-plots to assess whether the fitted negative binomial distribution represented the empirical data sufficiently to predict excessive drinking. This was done by predicting individual NABW values and comparing the quantiles with sample quantiles from the National Health Survey for the period 2014–2022. Fig 3 shows these qq-plots.

The (uncalibrated) fitted model predicts too many excessive drinkers. This is also shown by Table 1 "no calibration". One reason that the excessive drinking prevalence from the data and the model do not match is that there are relatively many persons who drink exactly 14 (women) or 21 (men) beverages per week, and fall just below the threshold of excessive drinking. Another explanation is that the data is more skewed and exhibit higher kurtosis compared what is captured by the negative binomial fit.

To obtain a distribution for the NABW that correctly predicts excessive drinking, the parameters of the negative binomial model were modified through a calibration procedure. In this procedure, the individual mean $\mu$ and dispersion parameter $\theta$ were scaled separately for men and women through scale parameters $\kappa_s^{ED}$ and $\lambda_s^{ED}$ (ED: excessive drinking, s: sex). Hence, the calibrated negative binomial distribution for the second submodel is given by

$$(NABW - 1) \sim NB\left(\kappa_s^{ED} \cdot \mu, \ \lambda_s^{ED} \cdot \theta\right), \tag{4}$$

where $\mu$ and $\theta$ still have the same value as in the uncalibrated model. The parameters $\kappa_s^{ED}$ and $\lambda_s^{ED}$ were estimated through a numerical optimization procedure. The squared relative differences between estimated and observed prevalences of excessive drinking were minimized, which is explained in S4 File. Trend breaks affected the shape of the

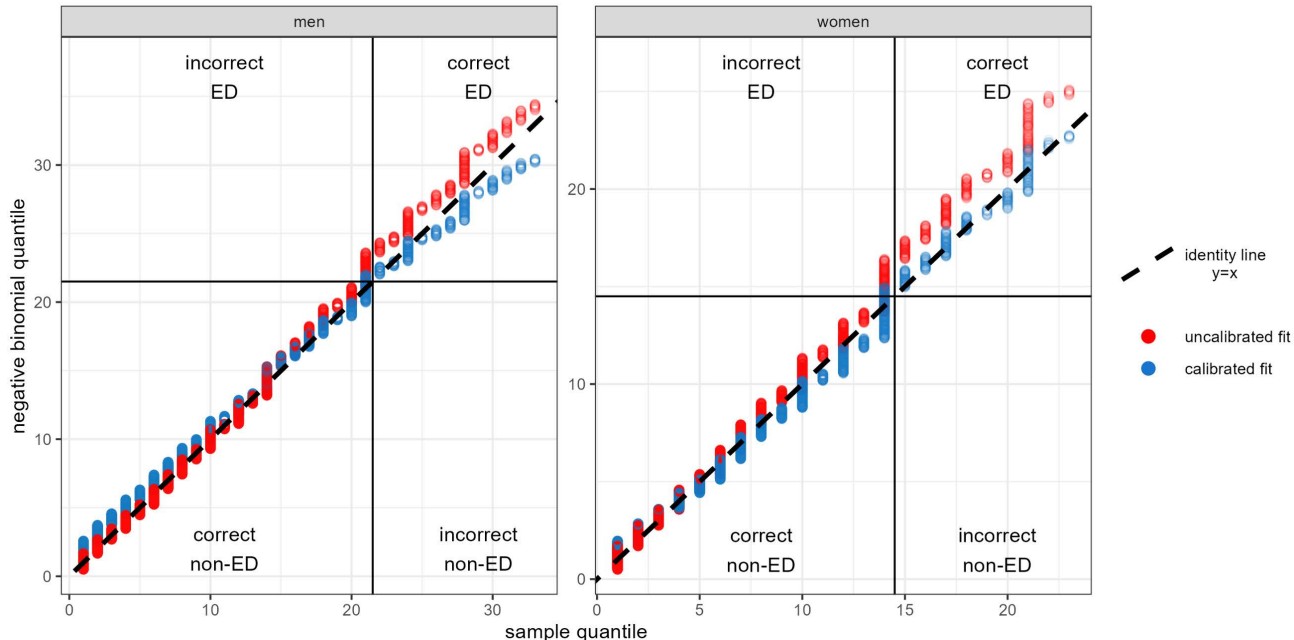

**Fig 3. Qq-plots for submodel 2.** Data sample quantiles versus predicted quantiles for both the uncalibrated fit (red) and the calibrated fit (blue), predicting NABW and excessive drinking (ED) for adult men and women in 2014-2022. Horizontal and vertical lines indicate excessive ED thresholds. These thresholds define four sections to distinguish the uncalibrated model's overestimation of excessive drinking ("incorrect ED"). Truncated for quantiles > 33 (men) and 23 (women), showing > 95% of the data. For visualisation purposes, we assumed the negative binomial quantiles to be uniformly distributed between integer values.

**Table 1. Results of the total model.**

| | | Total alcohol model | | |
|---|---|---|---|---|
| | *Data* | *No calibration* | *Calibrated for excessive drinking* | *Calibrated for Heavy drinking* |
| *Drinking [%]* | 74.2 | 74.1 | 74.1 | 74.1 |
| *Mean NABW (sd)* | 6.9 (11.0)[a] | 6.9 (9.8) | 6.3 (8.4) | 6.5 (8.7) |
| *Excessive drinking [%]* | **8.5** | 10.1 | **8.6** | N/A |
| *Heavy drinking [%]* | **8.8** | 10.0 | N/A | **8.9** |

Drinking prevalence, mean and standard deviation of the NABW and excessive and heavy drinking prevalences for adults according to data and total alcohol model predictions (not calibrated and calibrated), weighted mean over the calibration period (2014–2022). Calibration targets and corresponding estimations highlighted in **bold**.

[a]For heavy drinking, the mean NABW (sd) in the data is 6.5 (9.8) due to removal of inconsistent data.

distribution of the NABW. Therefore, the target of the calibration procedure was the mean observed prevalence between 2014 and 2022, which was the most recent period without trend breaks.

## Submodel 3: Heavy drinking

The distribution of the NABW contains information about the amount of alcohol that is consumed in a typical week. To predict heavy drinking, additional information is needed on how the alcohol consumption is spread over the days in the week. This information is contained (implicitly) in the third submodel for heavy drinking. We defined the probability for an

individual to be a heavy drinker as: $p^{hd}$, and parameterized the corresponding log-odds, for an individual with sex $s$, level of education $e$, age $a$ and reported alcohol amount $NABW$, at calendar time $t$ as follows:

$$\log\left(\frac{p^{hd}}{1 - p^{hd}}\right) = \beta_0 + (\textit{main effects}) + (\textit{NABW interactions}) + (\textit{age interactions}) + (\textit{trend break term}) \tag{5}$$

where:

$$(\textit{main effects}) = \beta_1 \cdot t + \beta_2 \cdot D^s_{female} + \beta_3 \cdot D^e_{middle} + \beta_4 \cdot D^e_{high} + \beta_5 \cdot a + \beta_6 \cdot a^2 + \beta_7 \cdot a^3 + \beta_8 \cdot NABW \tag{6}$$

$$(\textit{NABW interactions}) = NABW \cdot (\beta_9 \cdot t + \beta_9 \cdot D^s_{female} + \beta_{10} \cdot D^e_{middle} + \beta_{11} \cdot D^e_{high} + \beta_{12} \cdot a) \tag{7}$$

$$(\textit{age interactions}) = a \cdot (\beta_{13} \cdot t + \beta_{14} \cdot D^s_{female} + \beta_{15} \cdot D^e_{middle} + \beta_{16} \cdot D^e_{high}) + \beta_{17} \cdot a^2 \cdot t \tag{8}$$

$$(\textit{trend break term}) = H(t - 2010) \cdot (\beta_{19} + \beta_{20} \cdot D^s_{female}) + H(t - 2012) \cdot (\beta_{21} + \beta_{22} \cdot D^s_{female})$$
$$+ H(t - 2014) \cdot (\beta_{23} + \beta_{24} \cdot D^s_{female}) \tag{9}$$

In Equations 6–9, $D$ denotes a dummy variable (e.g. $D^e_{middle}$: 1 if middle education, 0 otherwise), and $H(\tau)$ the Heavyside step function. For this model, variables were included based on theoretical reasons and p-values. That is, according to the definition, heavy drinkers consume at least 4 (women) or 6 (men) beverages per week (i.e., at least during one day of the week), so we expected the NABW to be an important predictor for heavy drinking. Therefore, this covariate and its two-way interactions with calendar time, age, sex and education, were included (Equations 6 and 7). Age was parameterised using ordinary polynomial functions (degree chosen based on p-values<0.05). Main effects for age, sex and education were included, as well as a main effect for calendar time to capture the period trend (Equation 6). We allowed the age effect to differ by calendar time, sex and education by including age interactions (Equation 8). Interactions of the trend break terms with sex were included to capture the change in the definition of heavy drinking for women in 2012 (Equation 9). For consistency, we added sex interactions for all trend break periods.

For some individuals in the data, the reported NABW was inconsistent with their reported heavy drinking status. 0.4% of the individuals is no heavy drinker according to the corresponding heavy drinking question, whilst this is impossible according to their NABW (for men: $\geq 42$ (6 beverages per day 7 days per week) and for women: $\geq 28$ (4 beverages per day 7 days per week)). These individuals were removed from the dataset for this submodel, as these outliers negatively affected the fit of the logistic curve based on the NABW. All parameters were estimated through logistic regression.

**Calibration.** We estimated the heavy drinking prevalence with the total alcohol model, where the negative binomial distribution from the second submodel, calculating the NABW, serves as input for the heavy drinking submodel (see Fig 1). The total alcohol model overestimated the heavy drinking prevalence (see Table 1 "no calibration"), because the negative binomial distribution was not flexible enough to reflect the data. Therefore, we calibrated the negative binomial distribution from the second submodel with the goal to adequately predict heavy drinking. As explained above, a calibration procedure minimized differences between the estimated and mean observed prevalence of heavy drinking between 2014–2022 by estimating scaling parameters $\kappa^{HD}_s$ and $\lambda^{HD}_s$ (HD: heavy drinking, s: sex) (see S4 File).

## Validation of the total model

The total model predicts the NABW and excessive and heavy drinking based on the combined submodels in three stages (see Fig 1). Using demographic information of adults from the National Health Survey, all stages were consecutively

applied to predict the drinking status and/or NABW, excessive drinking status and heavy drinking status for individuals in each year. Subsequently, population-level model predictions were calculated and compared to the data, performing within sample validation for the most recent period without trend breaks (2014 until 2022).

## Results

The alcohol model provides the functions and parameter estimates for submodels for drinking, the NABW and excessive drinking, heavy drinking. All parameter values are listed in Tables A-C in S5 File.

The first submodel (model fit depicted in Fig A in S6 File) accurately predicted the drinking prevalence on a population level, judged from the predicted overall prevalence (see Fig 3 for NABW = 0 and Table 1). Fig 4 shows the mean (including time trend) and percentiles of the estimated distribution, by sex and education, for NABW predictions of submodel 2. The highest alcohol consumption occurs around the age of 25 across all education and sex groups, except for men with a lower education. The consumption decreases until the age of about 35. After the age of 35 the alcohol consumption increases (again) towards the age of approximately 70. After age 70, alcohol consumption decreases. In general, women consume fewer alcoholic beverages per week than men. Also, the variance of the distribution is smaller for women. In general, the NABW decreases over time. Fig 2 shows the frequency distribution of the NABW for men and women according to data, and from the predictions of the uncalibrated and calibrated model. The figure suggests a proper fit to the data.

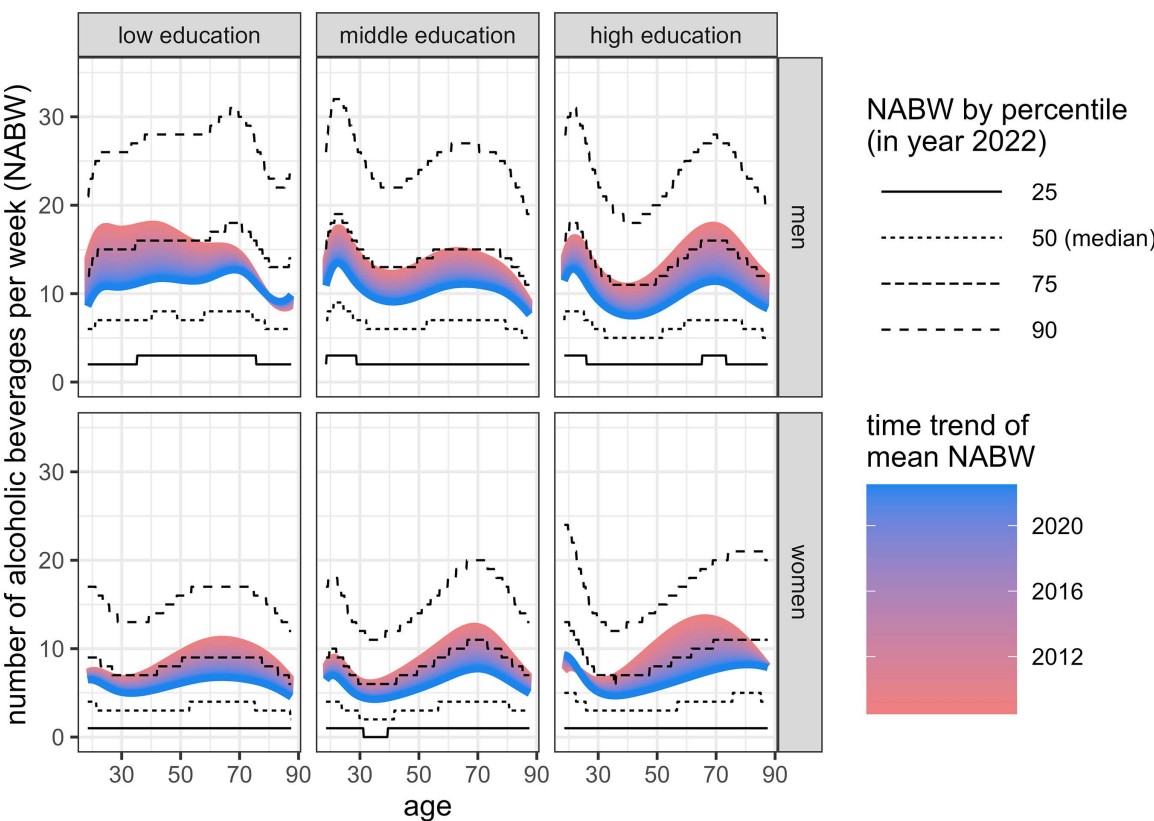

**Fig 4. Submodel 2 fit.** Uncalibrated negative binomial distribution predicting the NABW for adult drinkers (submodel 2) – four percentiles in year 2022 representing the spread of the distribution and colour gradient representing calendar time effect (time trend – utter red: 2008, utter blue: 2022) of the mean NABW, over age, by sex and education.

Looking back at the qq plot in Fig 3, it shows that at the thresholds of excessive drinking, the calibrated fit follows the identity line more closely compared to the uncalibrated fit for men and women. The calibration procedure yields a scaled negative binomial distribution with a higher density under the excessive drinking thresholds and a lower density above the excessive drinking thresholds. This means that, for individuals, more NABW values below the excessive drinking threshold are predicted. As the uncalibrated distribution predicts too many excessive drinkers, this results in a better prediction of excessive drinking prevalences on a population level (Table 1). Overall, the calibrated distribution shows no large deviations from the identity line (Fig 3), and from the data in the histogram (Fig 2).

Fig 5 shows the predictions from submodel 3, calculating the probability of being a heavy drinker, for a range of NABW values. As expected, the heavy drinking probability increases with the NABW. The conditional probability of heavy drinking, given the NABW, decreases as age increases. The dependence on education is more modest. The conditional probability of heavy drinking is increasing over calendar time in all subgroups, although the time trend effect was smaller for higher NABW values. Over the 15 years, it has become more likely that, for a fixed NABW, people consume at least 4 (women) or 6 (men) of the weekly number of beverages on at least one occasion. This is especially the case for persons who drink fewer alcoholic beverages per week.

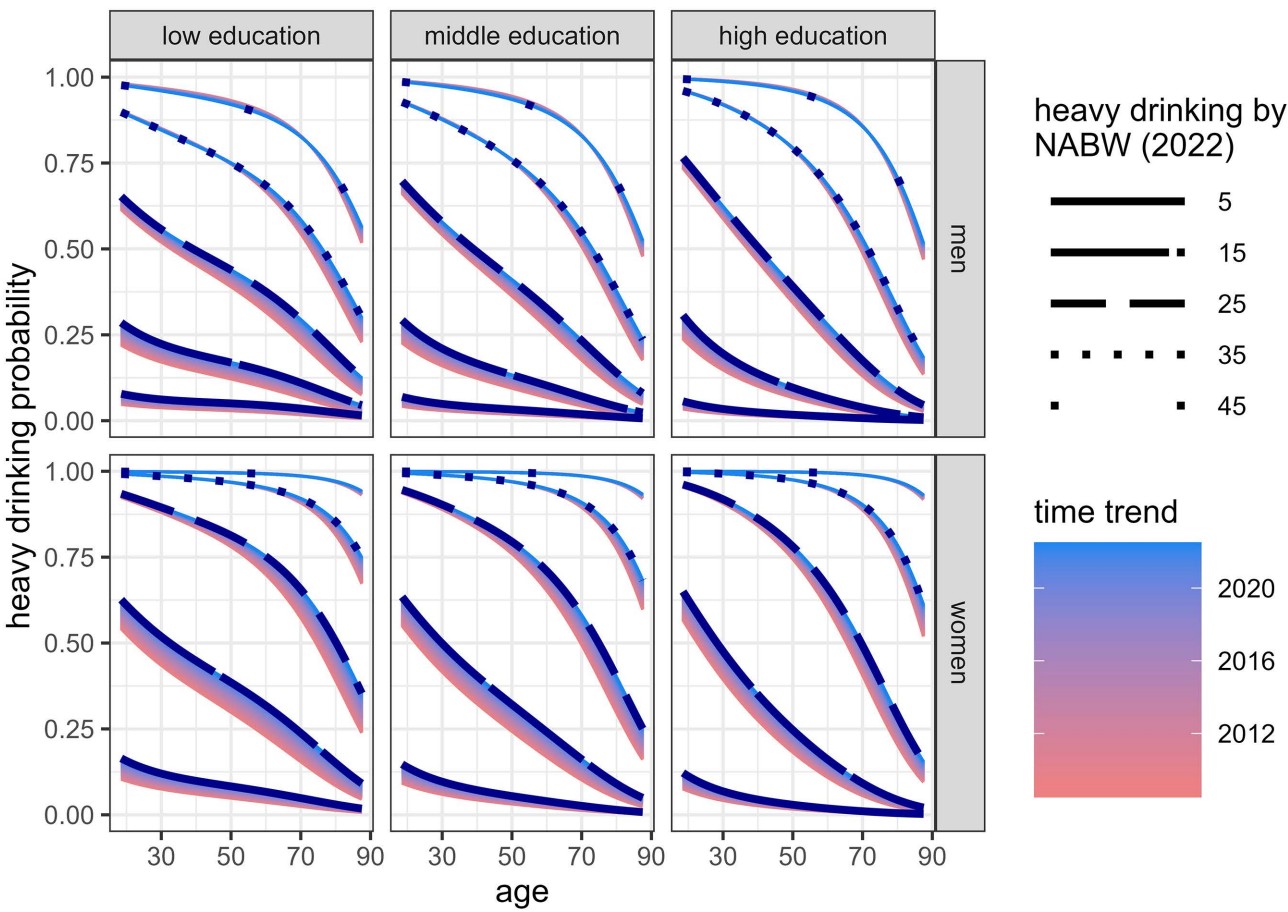

**Fig 5. Submodel 3 fit.** Predicted heavy drinking probability for adults (third submodel) – five values of the NABW over age and by sex and education. Black (dotted) lines indicate the heavy drinking probability by NABW in 2022. Colour gradients represent the time trends (utter red: 2008, utter blue: 2022).

The total alcohol model combines the three submodels. Table 1 shows (from the data and from the predictions of the total alcohol model, with and without calibration) the prevalences of drinking, excessive drinking and heavy drinking, and the mean and standard deviation of the NABW. As expected, the calibration procedures for excessive and heavy drinking improved the estimation of respectively the excessive drinking (post-calibration error <0.2 percent point) and heavy drinking prevalence (post-calibration error <0.1 percentage point). This was at the expense of the estimates for the mean NABW in the excessive drinking calibration (excessive drinking: −8.5%; heavy drinking: < 0.1%) and, in particular, the standard deviations of the calibrated NABW distributions (excessive drinking: −30.8%; heavy drinking: −12.5%). Fig 6 shows the predictions of the total alcohol model for excessive and heavy drinking over calendar time in the years 2014–2022 against observed data, indicating an acceptable model fit.

## Discussion and conclusion

We constructed an alcohol model that predicts, in three stages, (1) the probability of drinking alcohol, (2) the number of alcoholic beverages per week (NABW) and excessive drinking, and (3) the probability of being a heavy drinker. The model takes into account essential components to predict future alcohol consumption, such as historical time trends in alcohol consumption and demographic aspects. By estimating a discrete distribution for the NABW and allowing calculation of population-level prevalences of excessive and heavy drinking, our model can translate the impact of policy interventions, that may result in changes in the NABW, into the impact on excessive and heavy drinking. This enables the evaluation of goals to reduce excessive and heavy drinking, such as those set by the Dutch government [8].

A comparison between within-sample predictions of the total alcohol model and the observed excessive and heavy drinking prevalences over the years 2014–2022 shows that, after calibration, the model indeed successfully predicts the excessive and heavy drinking prevalences, based on the NABW (errors <0.2 pp). This was at the expense of estimates for the mean (errors < 9%) and, in particular, the standard deviation (errors <31%) of the NABW. For predictions of the mean

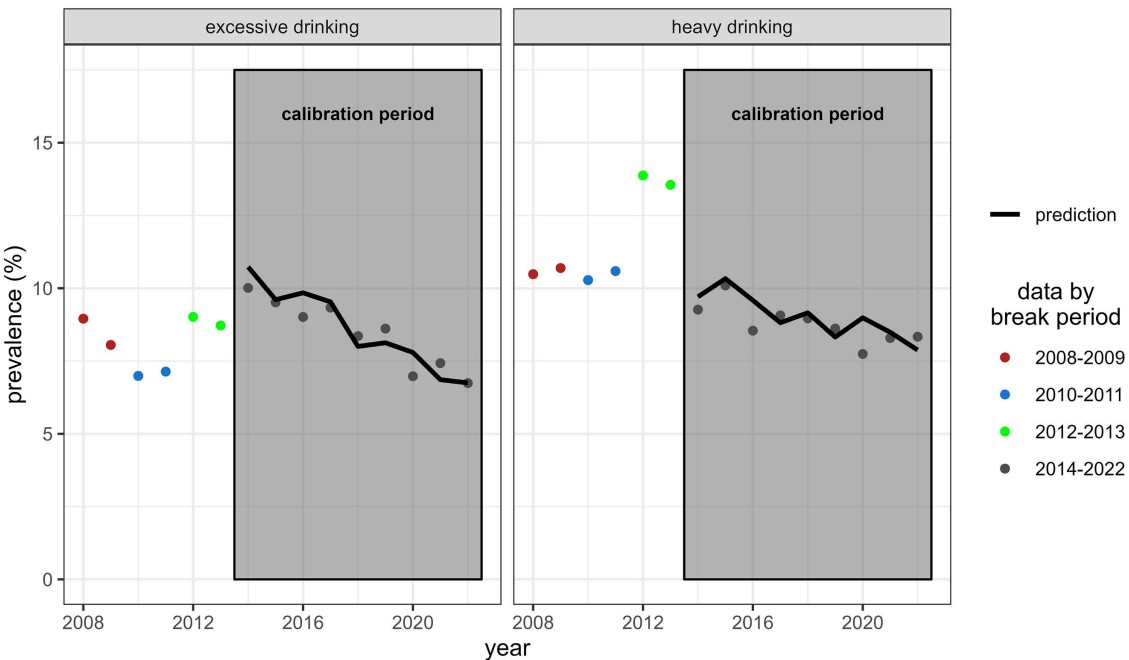

**Fig 6. Total model predictions.** Prevalences of excessive and heavy drinking over time for adults, as predicted by the total alcohol model in the calibration period (2014-2022), and from data for the four trend break periods.

NABW, the uncalibrated alcohol model should be used, which provides good estimates of the mean, and better estimates of the standard deviation, compared to the calibrated model (see Table 1).

In modelling excessive and heavy drinking, calibration was needed due to the (irregular) distribution of the NABW in the data, which could not be reflected by the negative binomial distribution. Alternative approaches exist that could potentially deal with the irregular NABW response variable. For example, Berger and Tutz [37] proposed a semiparametric model that allows the data itself to determine the distribution of the response variable. This approach is more flexible, and might capture some of the irregularity of the empirical distribution of the NABW, resulting in more accurate prediction of excessive and heavy drinking in the total alcohol model. However, a parametric distribution was preferred, since capturing all irregularity of the NABW response variable in the NABW submodel may bias estimates of intervention effects on excessive drinking. This is because in the empirical data, due to the underlying elicitation, there are relatively few people drinking 1 or 2 beverages more than the excessive drinking threshold (NABW of 15 or 16 for women and NABW of 22 or 23 for men). In practice, intervention effects sizes are generally below 2 beverages [19–21]. Consequently, intervention effects will be underestimated when using a non-smooth (semi-parametric) distribution capturing these data irregularities (assuming that the actual, unbiased, population distribution of the NABW is smooth (or: not irregular)). Hence, smoothening of the NABW response variable through our parametric approach (see Fig 2) is essential when accounting for the model's goal of predicting intervention effects. Combining this approach with our calibration procedure ensures the model's capability of successfully predicting excessive and heavy drinking prevalences. Qq-plots showed that the calibrated NABW distribution properly reflects the data, especially near excessive drinking thresholds.

The model has some limitations with regard to the data on which the model was fit. The data contains missing values, and these records were removed for fitting each submodel. From 2010 until 2013 there was a relatively large group of missing values in the alcohol items of the survey (+/-45%) in contrast to the other years (<5%). As our model already accounts for a trend break in this period, and accounts for (and thus imputes based on) age, sex and education, we did not further impute these missing data. This may induce a bias to the model if the missing values are not related to age, sex or education. Next, we removed inconsistent data for the heavy drinking submodel. Although the inconsistent data accounted for only a small fraction (0.4%) of the total, removing them may have induced selection bias. Lastly, as in many modelling studies using observational data, the model may reflect other biases from the data, such as response bias. Response bias (underreporting) is known to affect health survey estimates of alcohol consumption, as the alcohol consumption and the prevalence of heavy episodic drinking of non-responders is higher than that of responders [38].

We constructed an model to predict alcohol consumption patterns based on Dutch data. The model can be generalized by fitting (elements of) the model to health survey data from other countries, being generally available [39]. Although countries may use different terminology and definitions for their alcohol consumption patterns, the model can be applied for similar indicators, notably by using different cut-off values for excessive or heavy episodic drinking. This alcohol model provides insights in trends in alcohol consumption patterns in the Netherlands. Moreover, it is well-suited for incorporation in a simulation model to estimate the impact of interventions, resulting in a change in alcohol consumption, on population-level excessive and heavy episodic drinking prevalences. Therefore, future work includes comparing alcohol use predictions, using a health simulation model, against data from years on which the alcohol model was not based. Next, model extensions will allow broader applications. Heavy drinking behaviour can be modelled more explicitly by recording the number of occasions per week that an individual exceeds the heavy drinking threshold, rather than our current practise of modelling heavy drinking as a dichotomous variable. This would make it possible to incorporate interventions on heavy drinking behaviour, regardless of NABW consumption. Another extension is to incorporate alcohol-related disease risks or mortality using a health simulation model. For this purpose, the alcohol model may need to distinguish between never-drinkers and ex-drinkers, for which longitudinal data is required [40]. However, this study's alcohol model can already be applied to monitor alcohol use in The Netherlands by evaluating (future) trends of excessive and heavy drinking, as well as by calculating the impact of interventions [41]. These issues are central to the Dutch government public health efforts.

## Supporting information

**S1 File. National Health Survey items.**
(DOCX)

**S2 File. Data baseline characteristics.**
(DOCX)

**S3 File. Drinking prevalence over time.**
(DOCX)

**S4 File. Calibration procedure.**
(DOCX)

**S5 File. Model parameters.**
(DOCX)

**S6 File. Results for submodel 1.**
(DOCX)

## Acknowledgments

The authors would like to thank Professor Ardine de Wit and Peter Paul Klein for providing feedback on the manuscript and Dr Katalin Katona for her help with estimating drinking probabilities.

## Author contributions

**Conceptualization:** Jasper ten Dam, A. Jeroen Rodenburg, Hendrik Koffijberg, Anoukh van Giessen.

**Data curation:** Jasper ten Dam, A. Jeroen Rodenburg.

**Formal analysis:** Jasper ten Dam, A. Jeroen Rodenburg.

**Funding acquisition:** Anoukh van Giessen.

**Methodology:** Jasper ten Dam, A. Jeroen Rodenburg, Hendrik Koffijberg, Talitha L. Feenstra, Anoukh van Giessen.

**Project administration:** Jasper ten Dam, Anoukh van Giessen.

**Supervision:** Hendrik Koffijberg, Talitha L. Feenstra, Anoukh van Giessen.

**Validation:** Jasper ten Dam.

**Visualization:** Jasper ten Dam.

**Writing – original draft:** Jasper ten Dam, Anoukh van Giessen.

**Writing – review & editing:** A. Jeroen Rodenburg, Hendrik Koffijberg, Talitha L. Feenstra, Anoukh van Giessen.

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
