## [Decision Letter · Decision Letter 0]

16 Jul 2025

PONE-D-25-31774Modelling Alcohol Consumption Patterns to Enable Policy Impact AssessmentPLOS ONE

Dear Dr. ten Dam,

Thank you for submitting your manuscript to PLOS ONE. After careful consideration, we feel that it has merit but does not fully meet PLOS ONE’s publication criteria as it currently stands. Therefore, we invite you to submit a revised version of the manuscript that addresses the points raised during the review process.

We look forward to receiving your revised manuscript.

Kind regards,

Pengpeng Ye

Academic Editor

PLOS ONE

Journal Requirements:

This study was funded by the Dutch Ministry of Health, Welfare and Sport and the National Institute for Public Health and Environment (RIVM).

3. Thank you for uploading your study's underlying data set. Unfortunately, the repository you have noted in your Data Availability statement does not qualify as an acceptable data repository according to PLOS's standards.

5. We notice that your supplementary figures are uploaded with the file type 'Figure'. Please amend the file type to 'Supporting Information'. Please ensure that each Supporting Information file has a legend listed in the manuscript after the references list.

6. Please remove all personal information, ensure that the data shared are in accordance with participant consent, and re-upload a fully anonymized data set.

Reviewers' comments:

Reviewer's Responses to Questions

**Comments to the Author**

1. Is the manuscript technically sound, and do the data support the conclusions?

Reviewer #1: Yes

Reviewer #2: Yes

Reviewer #3: Yes

2. Has the statistical analysis been performed appropriately and rigorously?

Reviewer #1: Yes

Reviewer #2: I Don't Know

Reviewer #3: Yes

3. Have the authors made all data underlying the findings in their manuscript fully available?

Reviewer #1: Yes

Reviewer #2: Yes

Reviewer #3: Yes

4. Is the manuscript presented in an intelligible fashion and written in standard English?

Reviewer #1: Yes

Reviewer #2: Yes

Reviewer #3: Yes

5. Review Comments to the Author

Reviewer #1: Introduction - Quantify the Dutch disease burden attributable to excessive and heavy drinking. Add statistics to address the urgency of policy evaluation.

Methods - The paper states the National Health Survey uses "weighting variables" it doesn’t include details. Add a table detailing weighting methodology and its effect on prevalence estimates.

Methods- consider adding Sensitivity Analysis for Trend Breaks.

Methods - The paper removes individuals with "inconsistent" heavy drinking status (0.4%). This introduces selection bias. Acknowledge it.

Outline data needs for real-world monitoring of model predictions. create an early warning system for policy adjustments that can flag when things start to go off course. Include clear thresholds so you know when it’s time to step in and adjust policies or ramp up interventions.

Reviewer #2: Thank you for the opportunity to review this paper. The authors have undertaken a modeling study to develop predictive models of alcohol consumption patterns including weekly consumption, excessive consumption, and heavy episodic drinking using a representative sample of the Dutch population. Below I include minor comments to help clarify some elements in the manuscript.

Lines 78:81: The sentence beginning “Models that predict…” appears to be missing a word as the sentence is difficult to follow.

There are several places in the manuscript where the authors refer to incorrect figures. For example, Line 135 includes a reference to Figure 2 as a histogram, but this is actually Figure 3.

Line 136: Please include the specific version of R used for the analysis as the software does update and functions are sometimes deprecated.

Reviewer #3: Thank you for the opportunity to review this manuscript. This study proposes a new alcohol consumption predictive model based on data from The Netherlands. This study would be beneficial for the policy measure. The manuscript was well written. The methods were well designed. There are a few small points that might improve the manuscript:

- Line 95: Please give examples for number of alcoholic beverages. Is this equal to standard drink? Are both a can of beer and a bottle of wine counted as one alcoholic beverage, even their alcohol contents are different?

- Line 106-107: Please clarify the levels of education used in this study. Which levels are consider low, middle and high?

- Line 107 “year”: I think that “year” would indicate survey year. If so, please add detail for clarity.

- Line 361-366: Please explain in more details on how the missing data was handled.

- These models include demographics as the predictors of the alcohol consumption. I would like the ask if authors think that the changes in alcohol policy would influence the alcohol consumption. As for future studies, is it possible to incorporate modifiable factors or the factors that can be influenced by alcohol policy in the models?

- Regarding the generalizability of the study, please give us insights on whether this model would be able to be used outside of the Netherlands. If it is possible, please give some details on the potential conditions.

6. PLOS authors have the option to publish the peer review history of their article (what does this mean?). If published, this will include your full peer review and any attached files.

Reviewer #1: **Yes: **Mohsin Raza

Reviewer #2: **Yes: **Naomi K. Greene

Reviewer #3: No

---

## [Author Response · Author response to Decision Letter 1]

29 Aug 2025

See "Response to reviewers.docx"

---

## [Decision Letter · Decision Letter 1]

4 Nov 2025

Modelling Alcohol Consumption Patterns to Enable Policy Impact Assessment

PONE-D-25-31774R1

Dear Dr. ten Dam,

We’re pleased to inform you that your manuscript has been judged scientifically suitable for publication and will be formally accepted for publication once it meets all outstanding technical requirements.

Kind regards,

Pengpeng Ye

Academic Editor

PLOS ONE

Additional Editor Comments (optional):

Reviewers' comments:

Reviewer's Responses to Questions

**Comments to the Author**

1. If the authors have adequately addressed your comments raised in a previous round of review and you feel that this manuscript is now acceptable for publication, you may indicate that here to bypass the “Comments to the Author” section, enter your conflict of interest statement in the “Confidential to Editor” section, and submit your "Accept" recommendation.

Reviewer #1: All comments have been addressed

Reviewer #3: All comments have been addressed

2. Is the manuscript technically sound, and do the data support the conclusions?

Reviewer #1: Yes

Reviewer #3: Yes

3. Has the statistical analysis been performed appropriately and rigorously?

Reviewer #1: Yes

Reviewer #3: Yes

4. Have the authors made all data underlying the findings in their manuscript fully available?

Reviewer #1: Yes

Reviewer #3: Yes

5. Is the manuscript presented in an intelligible fashion and written in standard English?

Reviewer #1: Yes

Reviewer #3: Yes

6. Review Comments to the Author

Reviewer #1: (No Response)

Reviewer #3: (No Response)

7. PLOS authors have the option to publish the peer review history of their article (what does this mean?). If published, this will include your full peer review and any attached files.

Reviewer #1: **Yes: **Mohsin Raza

Reviewer #3: No

---

## [Editor Report · Acceptance letter]

PONE-D-25-31774R1

PLOS ONE

Dear Dr. ten Dam,

I'm pleased to inform you that your manuscript has been deemed suitable for publication in PLOS ONE. Congratulations! Your manuscript is now being handed over to our production team.

Kind regards,

on behalf of

Dr. Pengpeng Ye

Academic Editor

PLOS ONE